**Data Availability Statement:** https://figshare.com/articles/dataset/conversation_in_elderly_people/13227800.

**Funding:** This work was supported by JSPS KAKENHI Grant Number JP18K18140 (MSA), JP17H05920 (MO), JP18KT0035 (MO),

# Scaling laws in natural conversations among elderly people

**Masato S. Abe** [ID]*, **Mihoko Otake-Matsuura** [ID]

Center for Advanced Intelligence Project, RIKEN, Chuo-ku, Tokyo, Japan

* masato.abe@riken.jp

## Abstract

Language is a result of brain function; thus, impairment in cognitive function can result in language disorders. Understanding the aging of brain functions in terms of language processing is crucial for modern aging societies. Previous studies have shown that language characteristics, such as verbal fluency, are associated with cognitive functions. However, the scaling laws in language in elderly people remain poorly understood. In the current study, we recorded large-scale data of one million words from group conversations among healthy elderly people and analyzed the relationship between spoken language and cognitive functions in terms of scaling laws, namely, Zipf's law and Heaps' law. We found that word patterns followed these scaling laws irrespective of cognitive function, and that the variations in Heaps' exponents were associated with cognitive function. Moreover, variations in Heaps' exponents were associated with the ratio of new words taken from the other participants' speech. These results indicate that the exponents of scaling laws in language are related to cognitive processes.

## Introduction

Language is the most sophisticated means of human communication; it allows abstract thoughts and underlies various social activities, ranging from daily conversations to cultural accumulation [1, 2]. Brain function is involved in the processing of complex language; thus, impairments in cognitive functions can result in language disorders [3, 4]. Understanding the aging of brain functions in terms of language processing is crucial for modern aging societies because mental health problems, such as dementia, in elderly people have a huge impact on their daily lives and cause significant medical and economic costs worldwide [5, 6].

Previous studies on the relationship between language and cognitive functions have validated the use of linguistic features, such as verbal fluency, lexicon, semantic associations, and temporal patterns in spontaneous speech, in distinguishing people with dementia from healthy ones [3, 4, 7–12]. However, treating or stopping the progression of AD is very difficult at present. Instead, understanding of mild cognitive impairment (MCI), which is defined as a state between normal mental health and dementia, has more important practical implications because the detection of MCI makes patients participate in treatment trials to delay or prevent

JP19H01138 (MO), JP20H05022 (MO),
JP20H05574 (MO).

**Competing interests:** The authors have declared
that no competing interests exist.

cognitive decline [6]. Thus, it is crucial to understand the relationship between cognitive functions and language characteristics in healthy people and those with MCI.

Some studies on normal aging or MCI have also focused on the aging of the structure of semantics in the lexicon [12, 13]. These studies suggest that quantitative characteristics of language may be useful for describing and predicting the cognitive functions of elderly people as well as to distinguish normal pathological aging. Moreover, with regards to developing practical applications, some methods for distinguishing MCI using machine learning with large-scale language data have been proposed [14–16]. It has become easier to record and analyze large-scale data on language, such as ordinary conversations, due to the development of devices and algorithms. Hence, we expect that extracting information about cognitive functions from large-scale data makes it possible to deeply understand cognitive functions and language. However, in many cases, machine learning algorithms based on large-scale data specialize in predicting outcomes (e.g., healthy/MCI), while keeping the content of the model in a black box. Thus, it is difficult to gain insight into the cognitive mechanisms that underpin language. It is important to describe the quantitative relationship between cognitive functions and language based on large-scale datasets in order to understand the aging of cognitive processes.

In general, scaling laws are the most fundamental statistics in large-scale data. Scaling laws, which are generally defined as $f(x) \sim x^\mu$, where $\mu$ is a scaling exponent (here, "$\sim$" means that the left side is proportional to the right side), are ubiquitous in natural phenomena [17]. Interestingly, they are observed widely in brain and behavioral phenomena, such as neural dynamics, decision-making, semantic memory, memory retrieval, cognition, movements, language, and social dynamics [18–26]. The most famous example can be seen in the word patterns of human language [18]. So far, previous studies on language patterns in corpus data from written texts or spoken language [18, 21, 27, 28] have found two main scaling laws, namely, Zipf's law and Heaps' law. Zipf's law states that the frequency of the appearance of words with rank $r$ follows a type of power-law distribution $P(r) \sim r^{-\alpha}$, suggesting that a large number of words are rarely used, while a small number of words are often used. Since it was reported that the exponent $\alpha$ is close to 1 in most cases [29], the most frequent word will appear twice as often as the second most frequent word, three times as often as the third most frequent word, and so on.

Heaps' law, denoted as $N \sim M^\beta$, indicates that the number of different words $N$ (i.e., types) sub-linearly increases as the number of words $M$ (i.e., tokens) increases [30]. Empirical studies on written or spoken language have estimated the exponent $\beta$ to be approximately 0.7 [27, 28]. It is important to note that that Zipf's and Heaps' laws are universal, regardless of language, even though the structure of human language is highly complex in terms of context or grammar [27–29].

Most studies on scaling laws have been conducted from statistical and theoretical standpoints [18, 27, 31, 32]. It is unclear whether the spoken language in elderly people with low cognitive functions follows these scaling relationships, and how the variations in these scaling relationships are related to cognitive function; although, Zipf's laws have been reported in children and adults, and in schizophrenia cases [33–35].

In the present study, we focused on the spoken language of elderly people including people with MCI and healthy ones under natural conditions, including ordinary conversations. The questions included i) whether natural spoken language in elderly people including MCI follows the scaling laws mentioned above; ii) if so, how are the variations in scaling laws related to cognitive scores. We show that elderly people exhibited scaling laws in natural conversations, and that a higher Heaps' exponent was associated with higher cognitive function scores.

We concluded that cognitive function underlies speaking new topics in terms of the scaling laws of language.

## Materials and methods

### Data collection

Conversations among the study participants were recorded in order to obtain data on spoken language from healthy elderly people. The participants were recruited from the Tokyo Silver Human Resources Center, and were healthy Japanese retired adults, who speak Japanese as their mother language. To limit participants to healthy people and those with MCI, the exclusion criteria were set as follows: dementia; neurological impairment; any disease or medication known to affect the central nervous system; and a score of less than 24 points on the Japanese Mini-Mental State Examination (MMSE-J), which is a common criterion for screening dementia [36]. Assessment and screening to check participant eligibility were conducted based on medical interviews, neuropsychological tests, and self-reported questionnaires. A total of 72 participants underwent screening, and 7 were removed. Thus, the sample size for our data was 65 (30 males and 35 females), with a mean age of 72.6 ± 3.2 (SD) (range, 66–81), and a MMSE-J score of 28.0 ± 1.46 (mean ± SD).

Cognitive function tests were conducted for each participant prior to recording the conversations. The tests included the Japanese version of the Montreal Cognitive Assessment (MoCA-J) [37], a logical memory test (I + II) in the Wechsler Memory Scale-Revised [38], and the digit symbol coding test and digit span (forward + backward) in the Wechsler Adult Intelligence Scale Third Edition (WAIS-III) [39]. MoCA-J was used to evaluate global cognitive function, and included a score related to educational history. The logical memory test I assess immediate recall of the content of a story as soon as the examiner has finished reading it, while logical memory test II assesses delayed recall 30 min later. The digit symbol coding test assesses the process speed and memory in digit symbol coding performance, which requires the subject to write down each corresponding symbol as fast as possible. The digit span (forward) assesses simple memory span, and the digit span (backward) assesses working memory capacity. These cognitive score values are listed in Table 1. It has shown that these cognitive scores for elderly people have high test-retest reliability. The correlation between two evaluations was 0.92 for MoCA, 0.75-0.99 for Logical memory I, 0.70-0.75 for Logical memory II, and 0.75-0.99 for Digit symbol coding [40, 41]. Note that the reliability of Digit Span is relatively low ($< 0.70$). The RIKEN Institutional Review Board approved this study, which was carried out in accordance with the ethical principles of the Declaration of Helsinki, and all participants provided written informed consent.

The recording of the conversations took place from June 2018 to September 2018 in an experimental room. The 65 participants were divided into 16 groups before the first recording; 15 groups had four participants and one group had five participants. The participation was based on the availability of the participants. Every week, the participants joined the conversations for approximately 30 min and for a total of 14 weeks. Therefore, we obtained

**Table 1. Cognitive scores.**

|  | MoCA-J | Logical memory | Digit symbol coding | Digit span |
|---|---|---|---|---|
| Mean ± SD | 25.7 ± 2.6 | 16.7 ± 6.9 | 53.8 ± 13.4 | 16.0 ± 3.0 |
| Factor loading of PC1 | 0.72 | 0.84 | 0.44 | 0.46 |

The cognitive scores were evaluated before the conversation experiments ($n = 65$). A larger value represents a higher performance.

approximately 7 hours of conversational data from each group. The group members were fixed in their initially allocated groups until the end of all the experiments; in other words, each participant had conversations with the same group members each week. The 16 groups were divided into free conversation conditions and discussion conditions. The participants in eight of the 16 groups talked with each other freely. The participants in the other eight groups made a short presentation on a predetermined theme (e.g., favorite places in the neighborhood), which was specified in advance each week, and included questions and answers sessions for participants within the same group. The latter is a previously developed method to prevent dementia [42]; however, in the current study, we focused not on the details and effect of the method, but on the conversational patterns extracted from the recorded conversational data.

## Conversational data pre-processing

Conversation transcriptions derived from the recorded audio data were quantitatively analyzed to investigate word production patterns. Google Cloud Speech-to-Text (Google, Mountain View, CA, 2018) was first applied to automatically transcribe audio to text data, and the text was manually checked by comparing it to the audio data and fixing any mistakes. Second, we automatically decomposed all text into words and performed lemmatization using MeCab (ver. 0.996), which is a useful tool for Japanese morphological analysis based on conditional random fields [43]. Finally, we obtained the data of the spoken words of each participant in all sessions, from the first to the fourteenth session, and analyzed the data using R (ver. 4.0.3). The datasets related to the words and cognitive scores are available at [44], and the R code for data analysis is in S1 Data.

## Cognitive function scores

We conducted a principal component analysis to summarize the four cognitive function scores of MoCA-J, WAIS III logical memory I + II (delayed), digit symbol coding, and digit span (forward + backward), as shown in Table 1. The first principal component (PC1) contained 40.6% of all variances, and the factor loading of each cognitive score on PC1 had the same sign (Table 1) because the four cognitive function scores positively correlated with each other. Therefore, we can conclude that the larger the value of PC1, the better the cognitive function. Hereafter, we use the PC1 value as the 'cognitive function score' in the main analysis. Note that for simplicity of interpretation, the cognitive score was normalized with mean = 0 and SD = 1. Although this value is easy to understand, we performed an additional analysis for the case for each original value in an additional analysis because the PC1 did not have so much variance as mentioned above.

## Zipf's law and Heaps' law

To quantitatively analyze word production patterns, we focused on scaling laws in language, which have been investigated in the context of statistical linguistics [18]. Previous studies have reported that most language data, including corpus data, robustly follow Zipf's law and Heaps' law [27, 29, 30, 45]. Zipf's law states that the relationship between the rank r of the number of words and the frequency $P(r)$ of words is described as $P(r) \sim r^{-\alpha}$; where $\alpha$ is the scaling exponent and has been reported to be approximately 1, and $\sim$ means that the left side is proportional to the right side. Heaps' law describes how the number of different words appear, and states that the relationship between the number of words $M$ and the number of different words $N$ follows a function $N \sim M^{\beta}$. In this study, we focused on whether words in the spoken language of healthy elderly people including MCI follow scaling laws, and on the variation of exponents in the scaling relationships if scaling relationships exist.

To fit the distribution to the data of the rank-frequency relationship, we compared six candidate distributions, including a power-law, shifted power-law, power-law with cut-off, and so on (see the details in S1 Appendix in S1 File). First, we fitted each candidate model to the data using the maximum likelihood estimation [46], and estimated the parameters (i.e., exponents) using the Nelder-Mead method for maximizing the log-likelihood. Then, we explored the best model using AICs and Akaike weights. With regards to Heaps' law, we used the least-squared method, estimated the scaling exponents, and calculated the R-squared value to evaluate goodness-of-fit.

## Results

### Zipf's law and Heaps' law in spoken language

We recorded conversations among healthy elderly participants ($n = 65$) and conducted tests to evaluate cognitive functions in advance of overall conversational experiments. The basic characteristics of the conversations among participants are shown in S1 Fig in S1 File. The mean of the total number of spoken words among the participants was $17,854 \pm 7,145$ (SD), and the mean of the different words spoken was $2,023 \pm 439$ (SD). Fig 1A shows the rank-frequency distribution of words, and Fig 1B shows the relationship between the number of words and the number of different words spoken by each participant, with lines colored according to the cognitive function scores. The rank-frequency distributions apparently follow a type of power-law distribution because the frequencies of words, except for higher rank ($r < 10 \sim$), seem to be on the line in the log-log plot, suggesting that the spoken words of elderly people follow Zipf's law. To verify this statistically, we fitted six candidate distributions, including a power-law distribution and a shifted power-law distribution, to our empirical data using rigorous statistical techniques (see Methods, S2 Appendix and S2 Fig in S1 File) [27, 46]. The results obtained by the model selection showed that all the rank-frequency distributions for the 65 participants

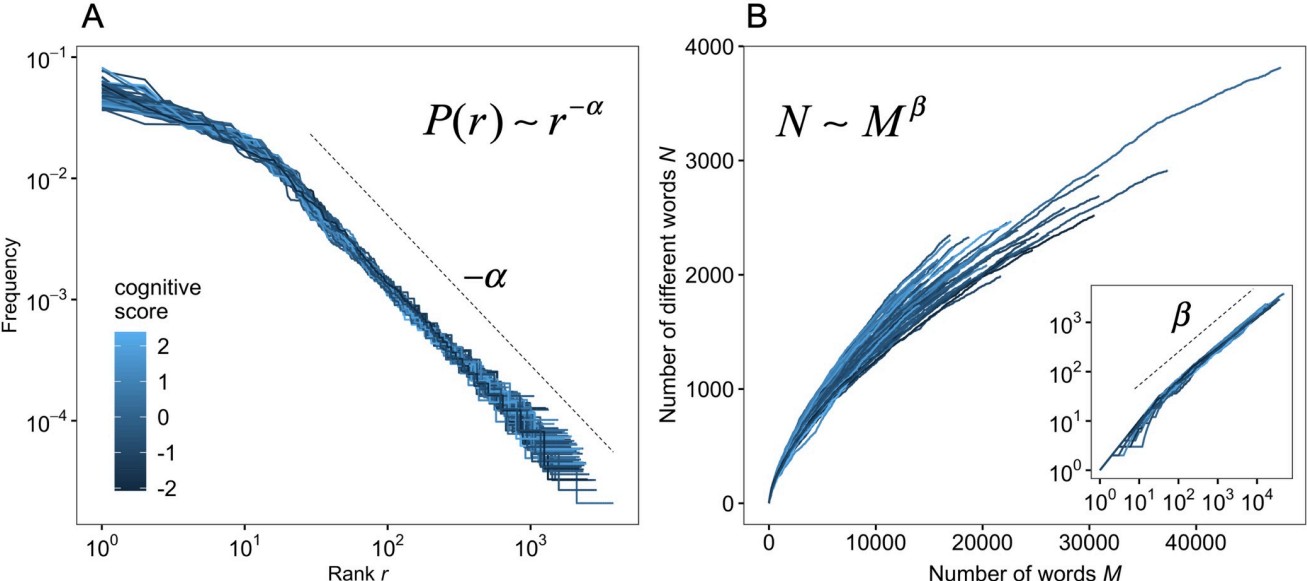

**Fig 1. Zipf's law and Heaps' law in natural conversations of elderly people.** (A) Zipf's law: Relationship between rank $r$ and frequency of words. The slope corresponds to the scaling exponent $-\alpha$. (B) Heaps' law: Relationship between the number of words $M$ and the number of different words $N$. The inset in (B) is a log-log plot of the main plot in (B), and the slope corresponds to the scaling exponent $\beta$. In (A) and (B), each line represents one participant, and the color represents the cognitive scores shown in (A).

were fitted to shifted power-law distributions (Akaike weights of shifted power-law distribution > 0.99). This statistical result suggests that spoken words in conversations among healthy elderly people follow Zipf's law, although the words in the higher rank (i.e., small $r$) do not follow a pure power-law distribution. This is consistent with previous studies, which have shown that written language follows a shifted power-law distribution [18, 29]. In the current study, the estimated Zipf's exponents $\alpha$ were $1.28 \pm 0.050$ (mean ± SD), and the range was (1.19–1.39).

We next analyzed the relationship between the number of words and the number of different words (i.e., newly-used words in each participant's conversation). It seems that the relationship follows Heaps' law (see inset in Fig 1B). Furthermore, there seems to be a relationship with slope = 1 between $N$ and $M$ at small $M$ (i.e., $M < m$, where $m$ is a break point), since most of the words are newly-used words in the vicinity of the initial states. Therefore, we fitted a double power-law model $N = M(M < m)$ and $N \sim M^{\beta}(M \geq m)$ to the data (S2 Fig in S1 File) [27, 47], estimated both the scaling exponent $\beta$ and threshold $m$ using an optimization method (Nelder-Mead method), and then calculated the R-squared values to evaluate the goodness-of-fit of the model to the data. We found that the Heaps' exponent $\beta$ was $0.671 \pm 0.02$ (mean ± SD), the range was (0.619, 0.736), and the average R-squared value was $0.999 \pm 0.001$ (SD) for an overall range of $M$ (S3 Fig in S1 File). This statistical analysis strongly indicates that the spoken words of elderly people also follow Heaps' law. Moreover, the mean of the estimated breakpoints was 29.

Although we independently analyzed Zipf's law and Heaps' law, it is known that the exponents of Zipf's law and Heaps' law are tightly dependent on each other as $\beta = 1/\alpha$ [27, 48, 49]. We found a significant negative correlation between $\alpha$ and $\beta$ in our results (Pearson's $r = -0.453$, $p = 0.0002$, Fig 2), which is qualitatively consistent with the theoretical result. Moreover, in the case of finite number of words, the exponent $\beta$ is lower than $1/\alpha$ as the number of words decreases [48]. Fig 2 also shows such a relationship in our empirical results.

## Relationship between word production patterns and cognitive scores

Next, we explored the relationships between the word production patterns, including the scaling laws estimated above, and the cognitive scores obtained from an independent cognitive test. Although one may expect that talkative people would have high cognitive scores, we found no significant correlation between the cognitive score and the total number of words spoken (Spearman's correlation coefficient $\rho = -0.11$, $p = 0.36$). We straightforwardly calculated a type-token ratio, which is defined as the value of the number of different words divided by the number of words. The Pearson's correlation coefficient between the ratio and the cognitive score was 0.19 ($p = 0.13$); thus, there was no significant relationship between the type-token ratio and cognitive scores. Then, we analyzed the relationship between the cognitive score and Zipf's law and found no significant relationship ($p = 0.077$) (Fig 3A, Table 2), using a linear mixed model with random effects (i.e., group).

We then examined the relationship between the exponent $\beta$ of Heaps' law and cognitive scores and found a significant relationship ($p = 0.002$) (Fig 3B, Table 2). The participants with higher cognitive scores were likely to have word patterns with a higher Heaps' exponent $\beta$, and vice versa. In contrast, the type of conversation and the age of participants showed no association with the exponent ($p = 0.94$ and $p = 0.93$, respectively) (Table 2). We confirmed a robust relationship between the exponent $\beta$ and all original cognitive scores, except for the digit span (Table 3). Thus, these results indicate that the variation in Heaps' law could be associated with the difference in cognitive functions. As an additional analysis, we also analyzed the data

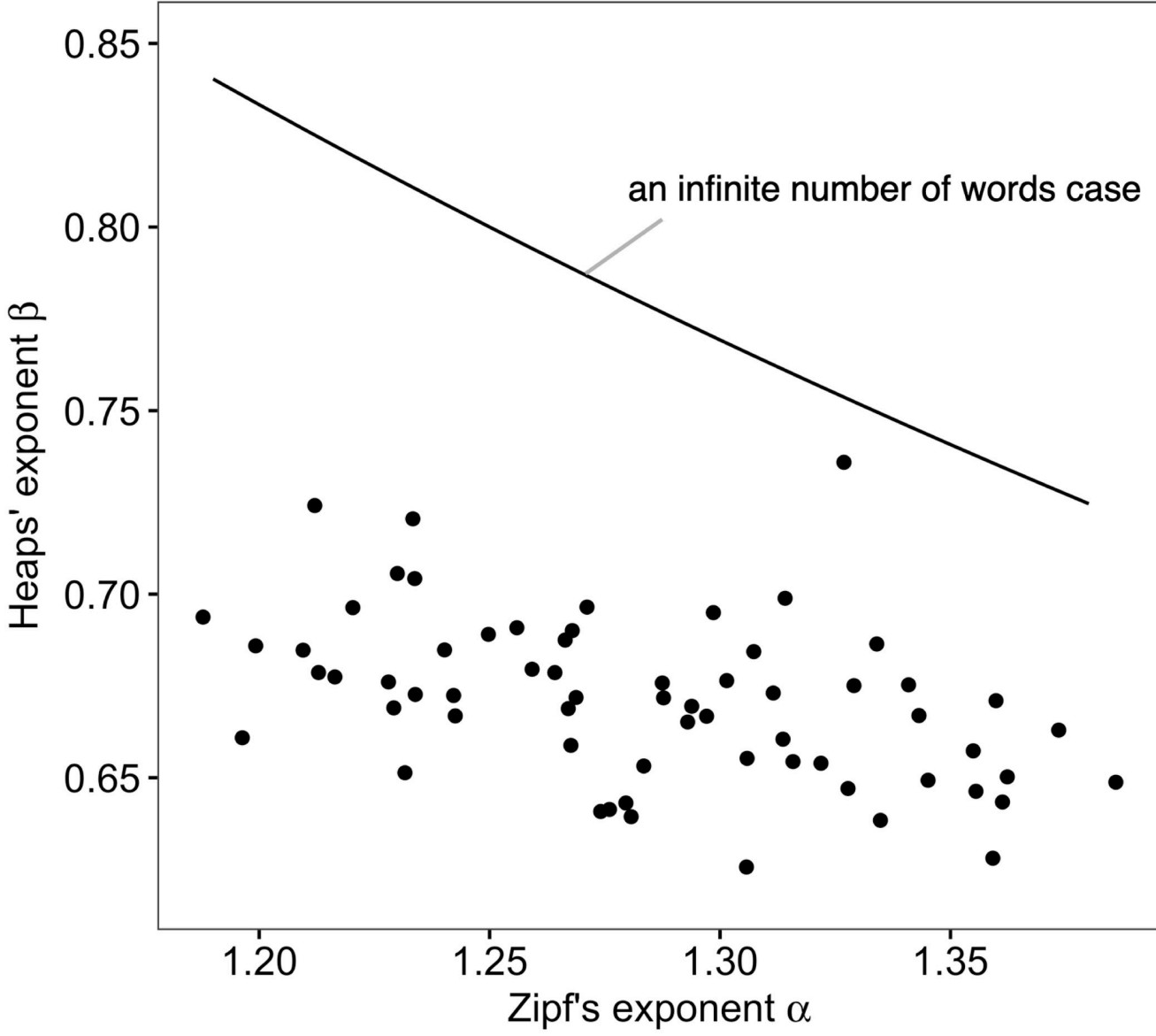

**Fig 2. Relationship between the estimated exponents.** The black dots represent the estimated exponents for each participant ($n = 65$). The solid line is $\beta = 1/\alpha$, which is the relationship between $\alpha$ and $\beta$ for the case of an infinite number of words.

without the prefixes because each participant may have a different prefix length (S3 Appendix in S1 File). The results were quite similar to the main results.

Since the number of words can affect the scaling exponent [48, 50, 51], it is possible that the observed relationship between the scaling exponents and the cognitive scores come from a bias effect by a different number of words. To remove such a bias, we investigated the relationship between the exponents and cognitive scores by using a fixed number of words $N_f$. In the analysis, we used data from the first word in the first session up to the $N_f$-th word in a session. We calculated the correlation coefficient between the exponents obtained from the analysis with $N_f$ and cognitive scores. Fig 4A shows that the correlation coefficient $r$ between the Zipf's exponent $\alpha$ and the cognitive score was not significant for most of range of $N_f$. In contrast, Fig 4B shows that when the data length was longer than approximately 10,000, the correlation

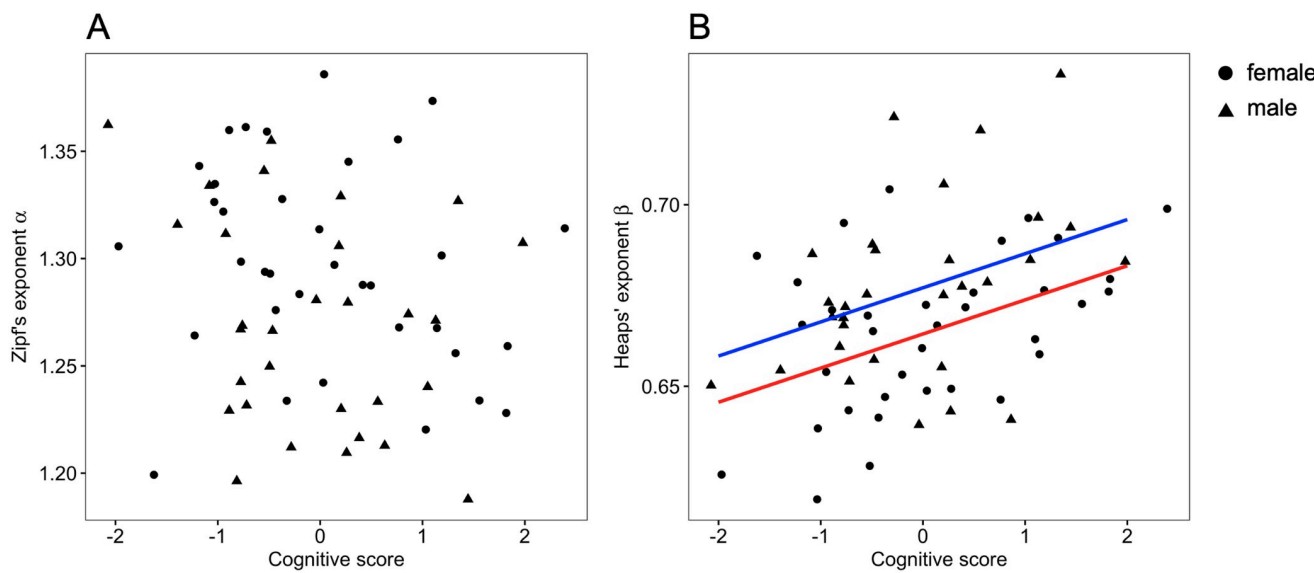

**Fig 3. Relationship between scaling laws and cognitive score.** (A) Cognitive scores vs. Zipf's exponent $\alpha$. Each point represents the data for each participant. (B) Cognitive scores vs. Heaps' exponent $\beta$. The blue (male) and red (female) lines represent the statistical model obtained from regressions.

**Table 2. Summary of linear mixed model on scaling exponents.**

|  | Explanatory variable: estimates (SE, p-value) | | | | |
|---|---|---|---|---|---|
|  | **Cognitive score** | **Gender (male)** | **Conversation type (presentation)** | **Age** | **Constant** |
| Zipf's exponent $\alpha$ | −0.011 (0.006, 0.077) | −0.029* (0.01, 0.015) | −0.010 (0.015, 0.512) | −0.007 (0.006, 0.236) | 1.303** (0.01, <0.001) |
| Heaps' exponent $\beta$ | 0.009** (0.003, 0.002) | 0.013* (0.005, 0.02) | 0.0005 (0.0055, 0.93) | 0.0002 (0.003, 0.94) | 0.664** (0.0043, <0.001) |

The linear mixed model had a random effect as a conversation group. The results were derived from all data sets. (* $p < 0.05$, ** $p < 0.01$)

coefficient $r$ between the Heaps' exponent $\beta$ and the cognitive score became significantly larger. This indicates that there was a clear relationship between the Heaps' exponent and cognitive scores even after removing the bias effect if only the number of words was larger than approximately ten thousand for each participant. Note that this amount of words corresponds to the vocal expressions of each participant for an hour or two in ordinary conversations.

## Source of newly-used words

The higher scaling exponent of Heaps' law could originate from the higher production rate of newly-used words [27]. Therefore, knowledge about the source of newly-used words could

**Table 3. Regression coefficients of each raw cognitive score for the scaling exponent $\beta$.**

| Cognitive score | Estimate (SE) | p-value |
|---|---|---|
| MoCA-J | 0.0038 (0.001) | 0.0006 |
| Logical memory (I + II) | 0.001 (0.0004) | 0.019 |
| Digit symbol coding | 0.0005 (0.0002) | 0.016 |
| Digit span (forward + backward) | −0.0009 (0.001) | 0.362 |

The linear mixed model had a random effect as a conversation group. The results were derived from all data sets.

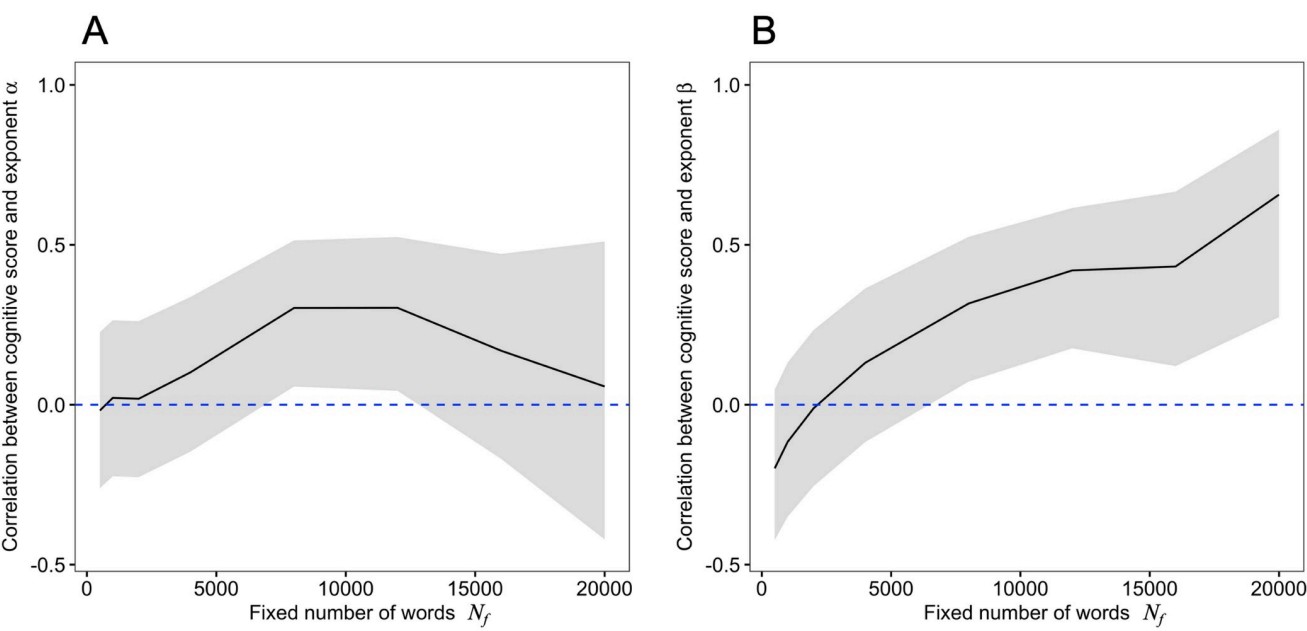

**Fig 4. Relationship between data length (fixed number of words) $N_f$ and correlation coefficients with Zipf's and Heaps' exponents.** The solid line represents Pearson's correlation coefficient $r$ between cognitive scores and the exponent $\alpha$ (A) and $\beta$ (B) when increasing the fixed number of words $N_f$. The shaded area indicates the 95% confidence interval, and the blue dashed line represents 0.

provide new insights into how people use different words [13]. To investigate this, we extracted the information on the source of origin of newly-used words, namely, from the participants' own internal memory or from other participants' verbal expressions during conversations. When a participant produced a newly-used word, we determined whether the words had already been used by the other group members by the time of the session. If the word had already been used, it was generally considered that the participant had heard the word through conversation, which suggests that the participant took the new information from others. If not, there is a high possibility that the word came from the participant's internal memory. For example, at the beginning, person A said that "I like apples", and then person B replied "I like apples, too!". In this case, person A produced a new word "apples" in the conversation, and person B used it. To quantify the source of newly-used words, we calculated the ratio of newly-used words from other participants based on the total number of newly-used words. Of note, this analysis was only conducted for the free conversation group, because the order of speaking in the presentation group was determined artificially. Fig 5 shows that the high exponents of Heaps' law are related to the high ratio of newly-used words from other participants ($r = 0.35$, $p = 0.047$), suggesting that a high susceptibility to the influence of other participants can provide a newly-used word production rate.

## Discussion

In this study, we quantitatively investigated the natural spoken language of healthy elderly people with various cognitive function scores including MCI from the viewpoint of scaling laws in word patterns. We also sought to explore the relationship between the scaling laws, including Zipf's law and Heaps' law, and cognitive function scores (Fig 1). We found that the scaling laws in spoken language were robust, irrespective of various cognitive function scores, from the result of fitting Zipf's law and Heaps' law. While we did not find a significant relationship between Zipf's exponent and the cognitive score (Fig 3A), the exponents of Heaps' law, that is,

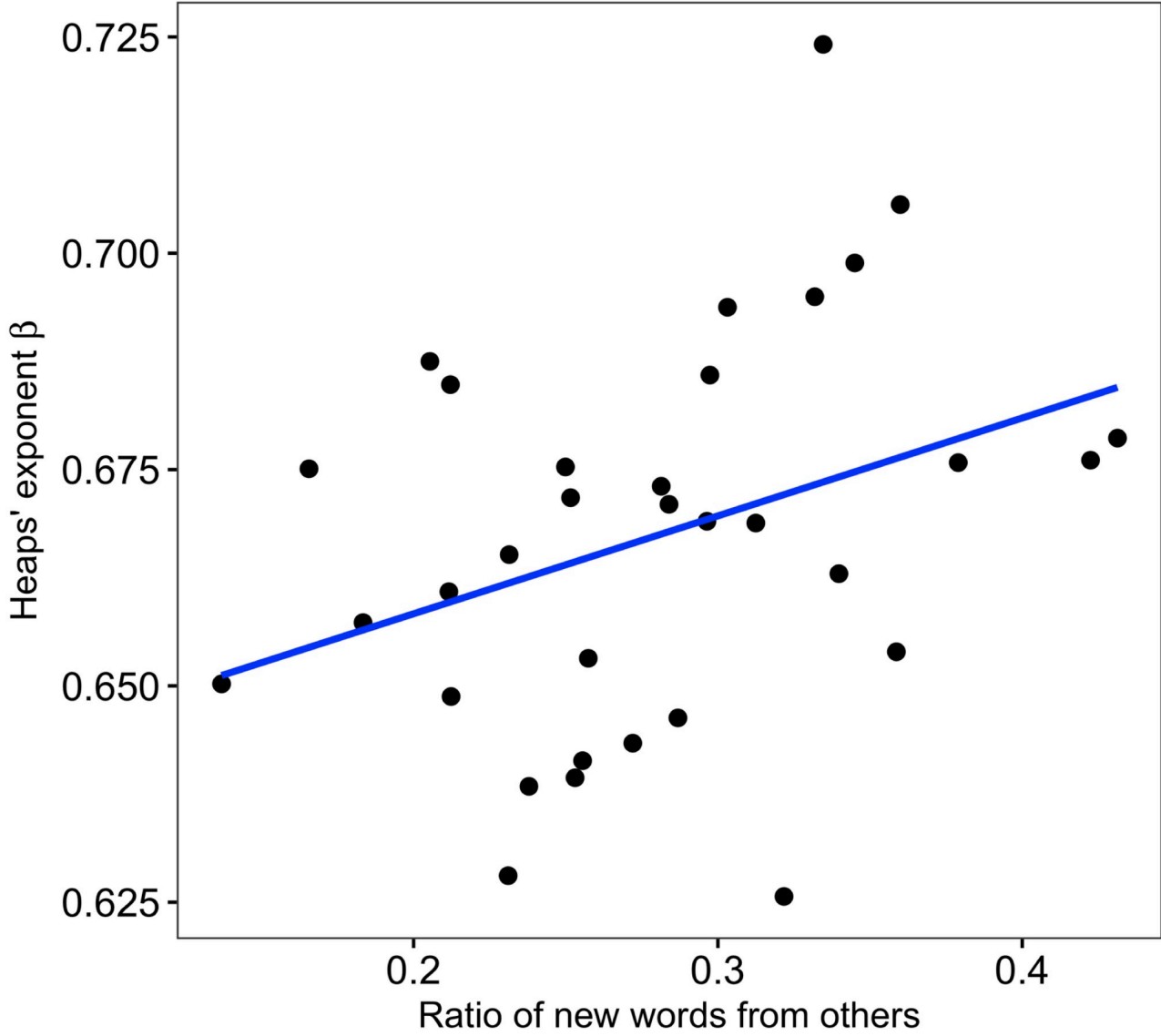

**Fig 5. Relationship between ratios of newly-used words from others and Heaps' exponents.** Each black dot represents each participant in free conversation groups. The horizontal axis is the ratios of newly-used words derived from other group members, and the vertical axis is the Heaps' exponents $\beta$. The solid blue line represents the linear regression line.

the slope of the relationship between the number of words and different words, were significantly associated with cognitive function scores (Fig 3B). It is worth noting that large $\beta$ corresponds to large vocabulary [27, 52]. Moreover, the large Heaps' exponents $\beta$ were related to obtaining different words from others and using them in conversations (Fig 5). Note that the participants were limited to elderly people, including MCI selected by screening based on an MMSE score $\geq$ 24. Therefore, the relationship between scaling laws and cognitive function may be fundamental knowledge in developing a method for early detection of cognitive impairments in healthy people.

In previous studies, it has been shown that Zipf's law and Heaps' law are related to each other, theoretically and empirically [27, 48, 49]. For an infinite number of words, the relationship between the two exponents is described as $\beta = 1/\alpha$ with a probabilistic model assumption

(Zipfian ensemble) [27] and without any probabilistic models [48]. Our results also show a similar relationship (Fig 2). Therefore, we expect that both exponents are related to cognitive functions. However, only Heaps' law was significantly associated with cognitive scores (Figs 3 and 4). Note that the $p$ value of the association with Zipf's law was $0.05 < p < 0.1$ (Table 2), and since the sign of the regression coefficient was negative, there might be a reasonable weak relationship between Zipf's exponent and cognitive scores. These facts suggest that high cognitive functions may lead to the usage of different words (i.e., Heaps' law), and that the process, such as how a word is selected from a set of already used words, may be less relevant to cognitive functions. Although there are some hypotheses as to whether Zipf's law or Heaps' law is the cause or the result [48], the above considerations suggest the possibility that Heaps's law arises from the cognitive process in word production without going through Zipf's law.

The usage of different words is important for communication and creating new ideas [13]. There are two main points regarding the mechanisms of different word usages. First, the number of different words reflects the participants' capacity to memorize, and particularly their capacity for long-term memory. Theoretically, it has been reported that Heaps' exponent $\beta$ is related to the size of the potential words [52]. Second, the number of different words suggests the ability of individuals to store and use new information, which could be crucial, especially for elderly people.

Our findings suggest that healthy elderly people with variations in cognitive scores still adhere to the scaling laws while one of the two exponents was significantly related to cognitive scores (Fig 1). This is similar to the result that the Zipf's exponents of patients with schizophrenia are different from those of healthy people [33]. Although such evidence has accumulated, the cognitive process that is related to producing these scaling laws and the control of the exponents remains unclear. Hence, future studies should include an analysis of whether the word patterns in patients with cognitive disorders including dementia or other mental disorders, follow the scaling laws, and how different their exponents are. Furthermore, a realistic computational model for word production needs to investigate the process underlying these scaling laws. This could assist in understanding why and how the scaling laws emerge, as could be useful for the development of early detection methods for cognitive disorders.

## Supporting information

**S1 Data. R code for analysis.**
(ZIP)

**S1 File.**
(PDF)

**S2 File.**
(TXT)

## Acknowledgments

We wish to thank Dr. Takuya Sekiguchi and Dr. Yukie Sano for the fruitful discussions. We also thank Dr. Seiki Tokunaga, Megumi Kubota, and Kaai Yamaguchi for their assistance with data collection.

## Author Contributions

**Conceptualization:** Masato S. Abe, Mihoko Otake-Matsuura.

**Data curation:** Masato S. Abe, Mihoko Otake-Matsuura.

**Formal analysis:** Masato S. Abe.

**Funding acquisition:** Masato S. Abe, Mihoko Otake-Matsuura.

**Investigation:** Masato S. Abe.

**Methodology:** Masato S. Abe.

**Writing – original draft:** Masato S. Abe.

**Writing – review & editing:** Masato S. Abe, Mihoko Otake-Matsuura.

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
