## [Decision Letter · Decision Letter 0]

17 Aug 2020

PONE-D-20-21971

Scaling laws in natural conversations among elderly people

PLOS ONE

Dear Dr. Abe,

Thank you for submitting your manuscript to PLOS ONE. After careful consideration, we feel that it has merit but does not fully meet PLOS ONE’s publication criteria as it currently stands. Therefore, we invite you to submit a revised version of the manuscript that addresses the points raised during the review process.

The referees recognize the interest and importance of your manuscript but indicated points that need to be addressed to ensure that the manuscript satisfies PLOS ONE publication criteria 3. 4. and 7.:

 3. Experiments, statistics, and other analyses are performed to a high technical standard and are described in sufficient detail.

4. Conclusions are presented in an appropriate fashion and are supported by the data.

7. The article adheres to appropriate reporting guidelines and community standards for data availability.

We look forward to receiving your revised manuscript.

Kind regards,

Eduardo G. Altmann

Academic Editor

PLOS ONE

Additional Editor Comments:

Please consider clarifying the following points around Eq. (1):

- Is n_c the same as N?

- Clarify why \\gamma is related to cognition, I understand the role it plays in the equation but it is unclear why this should be connected to cognition.

- it might be worth emphasizing that large \\beta corresponds to large vocabulary.

2. In your Methods section, please provide additional information about the demographic details of your participants. Please ensure you have provided sufficient details to replicate the analyses such as: a) a table of relevant demographic details and, b) a statement as to whether your sample can be considered representative of a larger population.

3. Please ensure you have thoroughly discussed any potential limitations of this study within the Discussion section.

6. Please include captions for your Supporting Information files at the end of your manuscript, and update any in-text citations to match accordingly. Please see our Supporting Information guidelines for more information: http://journals.plos.org/plosone/s/supporting-information

Reviewers' comments:

Reviewer's Responses to Questions

**Comments to the Author**

1. Is the manuscript technically sound, and do the data support the conclusions?

Reviewer #1: Partly

Reviewer #2: Partly

2. Has the statistical analysis been performed appropriately and rigorously? 

Reviewer #1: No

Reviewer #2: Yes

3. Have the authors made all data underlying the findings in their manuscript fully available?

Reviewer #1: No

Reviewer #2: No

4. Is the manuscript presented in an intelligible fashion and written in standard English?

Reviewer #1: Yes

Reviewer #2: Yes

5. Review Comments to the Author

Reviewer #1: This paper analyzes a set of data extracted from recorded conversations between mentally aged healthy Japanese-speakers. A cognitive score, which tests the cognitive capacity of an individual, is provided.

The goal of this paper is to find out:

1. Whether Zipf's and Heaps' laws are followed by the analyzed subjects.

2. Whether there was any relationship between the cognitive score for each subject and the values of the parameters computed for the said laws.

3. How the laws vary according to the cognitive score.

The paper suffers from many different problems, which I will try and explain in the following.

PRESENTATION.

In general terms, the presentation of the material is unclear.

Although this paper is not yet in production, the submission should follow some quality standards. Unfortunately, this is not the case. In fact, the submission contains numerous presentation errors, which resulted in a very uncomfortable review of the paper. I personally think that the paper should have been first rejected because of this and await a resubmission, just to fix its presentation problems. These are:

1. Figures appear at the end of the paper, out of their respective caption boxes, and in landscape format.

2. There are no hyperlinks in the text, which makes the reading still more uncomfortable.

3. No numeration of the sections. It is difficult to see the difference between sections and subsections.

4. The appendix is very short, it makes no sense to have it separately from the main article. It only makes the reading more painful.

The contents are the important part of the paper, because we all take the presentation part for granted. The presentation of in this submission is too poor, and it is a lack of respect for the journal and the reviewers alike.

INTRODUCTION.

In the Introduction section the authors do not clearly motivate their work. In the first paragraph they vaguely talk about the language and brain, then they talk about cognitive functions, and then they talk about the laws that are the backbone of the paper. The only motivation is in the lines 72-78 and are not sufficiently clear. They try to argue that finding out if these laws hold for *healthy* aged speakers, it may help to understand how language works with people with "low congnitive functions", but the paper is not related to that, from my point of view. I find the paper interesting by some of the results that it provides and also for what can be suggested by them, but the intorduction says nothing about it. Moreover, in the intro, there should be a few lines explaining what the main results of the paper are, and this is lacking as well.

DATA COLLECTION.

There are relevant details that are missing w.r.t. the processing of the data. You say that "we automatically decomposed all text into words using MeCab". Does it mean that you lemmatized the texts and kept the lemma only? For instance, in English, that would mean that the occurences of "book" and "books" would be transformed into the lemma "book". Is that what you performed, a lemmatization? Please clarify it.

RESULTS.

The methodology that the authors use in order to decide if the data follow the two laws (Zipf's and Heaps'), comparing them to seven candidate distributions is meaningful and follows standard practices: to fit the use likelihood + parameter estimation with nelder-mead + best model with AICs and Akaike.

In this section, the authors say: "The results obtained by the model selection showed that all the rank-frequency distributions for the 65 participants were fitted to shifted power-law distributions". However, there are no numerical results present in the paper. This reviewer has not been able to see this data on the paper, and if this is the case, this is an important flaw. These numbers *must* be on the paper, and also there should be a table that condenses all those results.

Figure S2.A and S2.B are Figures 1 and 2 averaged? I do not see what information is contained in Figures in S2 that is not already contained in Figures 1 and 2.

The graphs seem to show that there is a 2-states power-law (breakpoint at 10 un the x-axis) which seems to be consistent with state-of-the-art results. However, to say ""evidenced by the straight line in the log-log plot" is an error. Lines show, they do not evidence, prove nor confirm. The numerical values and statistical tests *do* show and prove. Unfortunately, they are missing in this paper. Unless shown, these results are meaningless.

Related with this discussion, I do not see why the authors did not use this same technique in order to verify that the Heaps' law is followed as well. That is: why don't you compare your data with different distribution as you did to see if Zipf's law was consistent. This is not what you do for Heaps' law (as stated in lines 219-229). Why this difference of methodology?

(As an aside comment, lines 219-229 are extreamly confusing and difficult to follow).

In Section "Relationship between word production patterns and cognitive scores" authors state that "we found no significant relationship between the cognitive score and the total number of words spoken (Spearman’s 237 correlation coefficient p = −0.11, p = 0.38)". I think that using a correlation test between a discrete variable (cognitive score) and a "continuous" variable (number of words) is an error. If it is not, please reference a paper where this technique has been used in a similar scenario (discrete vs continuous variable, where the discrete variable has only 5 possible values) Instead of that, there are some other techniques that could be used, as for instance, an Anova test between the alpha average and sd for score -2 and 2 (for instance) and then for scores -1 and 1, -2 and 0 etc... This test would say if the average value for the alpha (beta) values for each score are significantly different. A correlation does not seem to be convincing at all. It would help also to provide the average + sd for each cognitive score value in a graph. This comment should be extended to the rest of correlations that have been performed with the cognitive score variable and I give them low or little validity.

There is another issue with the relationship between the length of the conversations/productions and the rest of the analyses. First of all, lines 192-194 show that there is a large variability in both length and vocabulary. This means that there are speakers that speak more than others significantly. This induces a bias to the analysis. The authors claim that there is "no significant relationship between the cognitive score and the total number of words spoken" (the correlation is not significant). This seems to indicate that the results that are later obtanied relating the cognitive scores and the alpha/beta parameters are free of interference from the length. But this is not necessary true (taking aside the methodological problem that I just commented in the preceeding paragraph). In order to prove that the length (word production) is taken out of the equation (that is: length does not bias the analyses) one should make the analysis taking fixed-length samples and comparing the results obtanied in the same analyses with thoses presented in the paper. If the results were similar, then, it could be concluded that text length does not bias the results. Assuming that this effect may be "cancelled" by the fact that there is no relation between two of the variables is not necessarily true.

The authors also state that "the longer the data length (e.g., > 10,000), the higher the correlation coefficient (e.g., r close to 0.5) between the Heaps’ exponent and the cognitive score". That is: this implies that the longer the conversation, the more correlated the heaps exponent to the cognitive score, then, it implies that the length is relevant to this analysis, that is, the length *biases* the analysis. Said otherwise: we could have that it is the length of the text that provides a higher value of the parameter. This seems inconsistent with the above paragraph.

But moreover, this would not mean that "it is not necessary to analyze datasets with tens of thousands of words for each participant", as the authors point out. Maybe, a longer length would exhibit different results. It just means that for the lengths that are in the dataset, it is not relevant.

As for the section "Computational models bridging scaling laws and cognitive functions", I have not been able to understand the meaning of this section in this paper. The material contained here seems completly irrelevant to me, but I may be wrong. I would appreciate a reason why this section offers a new insight or adds new ideas. It seems that the authors tested a theoretical models, but I do not see any relationship with the rest of the paper.

In Section "Source of new words", gives me some deep theoretical doubts. The authors seem to suggest that there is a way to find out what words uttered by the speakers have been acquired during the conversations and which belonged already to the speaker's repertoire. However, it seems to me very unlikely to assume that old people acquire new words at all. Therefore, the assumption in this section seems seriously flawed to me: what is the evidence that this feature (word acquisition) is a productive feature in aged people? It would make perfectly sense in the case of a longitudinal study of children that are acquiring words, but in this case, it seems tremendously unclear and it has no theoretical support (at least, I see no reference in the text). If a subject utters a word in the conversation, it may be that this is because he heard it from a conversation partner. But from here, one can't imply that the subject *acquired* that word taking into account the age of the individuals analyzed (here I mean *acquire* in the linguistic sense of adding it to his own lexicon).

DISCUSSION.

In the Discussion section, the authors claim that "In contrast, MCI or dementia patients might not follow the scaling laws because repeating a certain word due to critical cognitive impairment or memory disorder may result in the collapse of the scaling laws". This is just a speculation that has no empirical evidence in the paper.

This section (Discussion) is a mix of different subjects with no meaningful narrative. I strongly suggest to rewrite this section, following a coherent line of discussion, not mixing future work (349-360), speculations with the discussion of the material contained on the paper. I would also avoid sentences like "we revealed the association between cognitive functions and word production patterns" because as this reviewer states above, this is debatable.

You say that "While we did not find a significant relationship between Zipf’s exponent and the cognitive score (Fig 2A), the exponents of Heaps’ law, that is, the slope of the relationship between the number of words and different words, were significantly associated with cognitive function scores". Taking aside that (as I previously commented on) the methodology that you use to assume that there is a relationship between the exponents of the Heaps' law and the congitive scores is flawed, the fact that this relation exists, but the relation between exponents of the Zipf's law and the same scores, how does it relate to the fact that both Zipf's and Heaps' laws are related? (see bibliographical suggestions at the end of this text). Is it consistent?

FINAL REMARKS.

The authors should discuss the relationship between both Zipf's and Heaps' laws. Some references that may be of help:

https://journals.plos.org/plosone/article?id=10.1371/journal.pone.0014139

https://doi.org/10.1076/jqul.8.3.165.4101

https://journals.aps.org/prl/abstract/10.1103/PhysRevLett.114.238701

In fact, along the paper, they seem to ignore that both laws are related, and that some of the results could be inconsistent with this assumption. That should also be discussed.

There is a lot to improve the paper, starting by the presentation. However, I would like to say that the material in this paper seems to be of interest. The study of Zipf's and Heaps' laws in diferent kind of people and languages, is an interesting topic of research.

But there is a lot to be improved: presentation, motivation, relevant methodological problems, a more systematic presentation of the results, stating *clearly* each of the relationships that have been studied, with what methodology, and present the *full* results (in the appendix if they are too extensive) with figures that help to understand them. Therefore, I would encourage the authors to improve the quality of their paper, be more consistent in their analyses.

Reviewer #2: The authors present a quantitative linguistics study of spoken Japanese conversation in healthy elderly subjects. The manuscript lays out several interesting results, but not all claims are properly sustained in my opinion. I believe the manuscript could be published if the following points are taken into consideration in a revised version:

1. I appreciate that the authors share the cognitive test results and fitted Heaps/Zipf’s exponents. However, for the sake of reproducibility, I would suggest to publish also the processed conversational data. While the fitting method used by the authors seemed robust enough to me, fitting of fat-tailed distributions is a delicate issue and many fitting procedures have been proposed in the literature. Therefore, I would be more comfortable if the reader is given access to the raw tokenized dataset, which would allow him/her to (i) verify the results of the manuscript and (ii) perform additional alternative analysis of the dataset. There are simple solutions to publish large datasets such as Zenodo.

2. The R code / necessary scripts to reproduce the results of the paper should also be published (as per PLOS editorial policy).

3. Is there a measurement error associated to the cognitive tests results? For instance, if each subject performed each test more than once, then it would be better to report the variances in addition to the averages. If each test was only taken once then it would be good to know, perhaps from the literature, what are the typical inter-subject variabilities of said tests.

4. I am not convinced the PCA-based cognitive score is a good metric for several reasons. First, it only captures 40% of the variability. My understanding is that the different cognitive tests measure different cognitive functions which could but do not necessarily correlate with each other. If that is the case, then taking the PCA of all measures might obscure more interesting results. Second, using the first principal component as a “summary” cognitive score has the undesirable consequence that scores of different subjects are not independent of each other anymore: to compute the score of one subject, we need the test results of all other subjects. I suggest the authors mention, at least, the loads of the first principal component, so that the reader has an idea of what are the weights of each test in the final cognitive score, and perform some additional analysis based on each cognitive test separately.

5. Zipf’s law and Heaps’ law are statistical laws tightly dependent on each other, see for instance (Lü, Zhang, and Zhou 2010; van Leijenhorst and van der Weide 2005; Font-Clos and Corral 2015, ). In this sense, the fact that only Heaps’ exponent significantly correlates with cognitive scores –but not Zipf’s exponent— might just be a technical artifact. Regardless, the number of tokens could certainly be a confounding variable in this case: First, there has been already ample debate in the community regarding the relation between text length and Zipf’s exponent, see (Corral and Font-Clos 2017; Bernhardsson, da Rocha, and Minnhagen 2009) (in summary, and depending on the fitting methodology etc, one tends to obtain larger exponents for longer texts). Second, having a quick look at the supplementary CSV data, it would seem that text length is a very good predictor of Heaps’ exponent, but not of cognitive score. The authors might want to attempt to more clearly untangle the role of text length in their analysis.

6. The reasoning of lines 269-272 is very unclear to me: basically, the Figure shows that in short datasets the association is lost, which would imply the opposite of what is being said? That is, we need to analyze large datasets to make sure we observe a meaningful association.

7. I do not see a clear connection between the computational model and brain cognitive functionality. Clearly, if Heaps’ exponent correlates with cognitive score, then this will also be captured by the model of Gerlach et al, but that does not bring new information per se. The authors say: “we focused on a parameter related to the decay rate of probability for new word production and interpreted it as a cognitive function.” So, the relation between word production (Heaps’ law) and cognitive function is an assumption of the authors, not a conclusion obtained from the analysis. This is clearly seen in Figure 4A (which would be obtained with any other dataset).

8. The authors claim “the scaling laws are useful to detect the tendency of cognitive decline, even in healthy people.”. I do not think this conclusion is well supported by the analysis presented in this manuscript.

9. The authors claim that “out approach requires only limited data from which to detect the relationship between cognitive functions and word patterns, even for healthy participants.”, at odds with the results presented in Figure 3.

Bernhardsson, Sebastian, Luis Enrique Correa da Rocha, and Petter Minnhagen. 2009. “The Meta Book and Size-Dependent Properties of Written Language.” New Journal of Physics 11 (12): 123015.

Corral, Álvaro, and Francesc Font-Clos. 2017. “Dependence of Exponents on Text Length versus Finite-Size Scaling for Word-Frequency Distributions.” Physical Review. E 96 (2–1): 022318.

Font-Clos, Francesc, and Álvaro Corral. 2015. “Log-Log Convexity of Type-Token Growth in Zipf’s Systems.” Physical Review Letters 114 (23): 238701.

Leijenhorst, D. C. van, and Th P. van der Weide. 2005. “A Formal Derivation of Heaps’ Law.” Information Sciences 170 (2): 263–72.

Lü, Linyuan, Zi-Ke Zhang, and Tao Zhou. 2010. “Zipf’s Law Leads to Heaps’ Law: Analyzing Their Relation in Finite-Size Systems.” PloS One 5 (12): e14139.

6. PLOS authors have the option to publish the peer review history of their article (what does this mean?). If published, this will include your full peer review and any attached files.

Reviewer #1: No

Reviewer #2: **Yes: **Francesc Font-Clos

---

## [Author Response · Author response to Decision Letter 0]

17 Nov 2020

Answers to Editor

-----

Please consider clarifying the following points around Eq. (1):

• Is nc the same as N?

• Clarify why gamma is related to cognition, I understand the role it plays in the equation but it is unclear why this should be connected to cognition.

• it might be worth emphasizing that large beta corresponds to large vocabulary. 

Response: Thank you for the helpful comments. However, following advice from our referee, we removed the computational model section. Nevertheless, we have emphasized that a large beta corresponds to a large vocabulary (lines 282-283).

Answers to Reviewer 1

In response to the valuable comments of the referee, I have modified my paper appropriately to address her/his concerns. Below I give a point-by-point reply to each criticism. Please note that the improved parts in the manuscript are indicated in blue.

----- Comment 1 -----

PRESENTATION.

In general terms, the presentation of the material is unclear. Although this paper is not yet in production, the submission should follow some quality standards. Unfortunately, this is not the case. In fact, the submission contains numerous presentation errors, which resulted in a very uncomfortable review of the paper. I personally think that the paper should have been first rejected because of this and await a resubmission, just to fix its presentation problems. These are:

1. Figures appear at the end of the paper, out of their respective caption boxes, and in landscape format.

2. There are no hyperlinks in the text, which makes the reading still more uncomfortable.

3. No numeration of the sections. It is diffcult to see the dierence between sections and subsections.

4. The appendix is very short, it makes no sense to have it separately from the main article. It only makes the reading more painful.

The contents are the important part of the paper, because we all take the presentation part for granted. The presentation of in this submission is too poor, and it is a lack of respect for the journal and the reviewers alike.

Response: We are sorry for making you uncomfortable. However, the format of the previous manuscript was according to the submission guidelines of PLOS ONE (https://journals.plos.org/plosone/s/submission-guidelines). The guidelines say that figures should be uploaded separately and and that they should appear at the end of the manuscript. In contrast, the captions should be written close to the citation of the figures within the main text. Also, the landscape formatting of the figures was performed by the automatic compilation of the submitted manuscript files during the PLOS ONE submission procedure. Therefore, we could not change it. In general, hyperlinks are not used in manuscripts written via Microsoft Word, which is different from LaTeX. The submission guidelines also state that section numbers should not be used. A short appendix is often used in empirical study papers. For example, an appendix sometimes includes only one supporting table or figure. To avoid these issues, we have rewritten the entire manuscript by using a LaTeX template for PLOS ONE. I expect that the present manuscript is easier to read than the previous one.

---- Comment 2 -----

INTRODUCTION.

In the Introduction section the authors do not clearly motivate their work. In the first para-

graph they vaguely talk about the language and brain, then they talk about cognitive functions,

and then they talk about the laws that are the backbone of the paper. The only motivation is

in the lines 72-78 and are not suffciently clear. They try to argue that finding out if these laws

hold for *healthy* aged speakers, it may help to understand how language works with people

with ”low congnitive functions”, but the paper is not related to that, from my point of view.

I find the paper interesting by some of the results that it provides and also for what can be

suggested by them, but the intorduction says nothing about it. Moreover, in the intro, there

should be a few lines explaining what the main results of the paper are, and this is lacking as

Response: Thank you for your helpful comments. We have improved the motivation of our study and added a few explanations for our main results in the Introduction (lines 10-19, 63-64, 67-70).

---- Comment 3 -----

DATA COLLECTION.

There are relevant details that are missing w.r.t. the processing of the data. You say that ”we automatically decomposed all text into words using MeCab”. Does it mean that you lemmatized the texts and kept the lemma only? For instance, in English, that would mean that the occurences of ”book” and ”books” would be transformed into the lemma ”book”. Is that what you performed, a lemmatization? Please clarify it.

Response: Thank your for your insightful comments. In the previous manuscript, we did not perform a lemmatization. Therefore, we conducted all analyses after performing a lemmatization. However, the results did not change qualitatively. We have adopted the new results in the present manuscript. We have added the explanation (lines 127-128).

---- Comment 4 -----

RESULTS.

The methodology that the authors use in order to decide if the data follow the two laws (Zipf’s and Heaps’), comparing them to seven candidate distributions is meaningful and follows standard practices: to fit the use likelihood + parameter estimation with nelder-mead + best model with AICs and Akaike.

In this section, the authors say: ”The results obtained by the model selection showed that all the rank-frequency distributions for the 65 participants were fitted to shifted power-law distributions”. However, there are no numerical results present in the paper. This reviewer has not been able to see this data on the paper, and if this is the case, this is an important flaw. These numbers *must* be on the paper, and also there should be a table that condenses all those results.

Figure S2.A and S2.B are Figures 1 and 2 averaged? I do not see what information is contained in Figures in S2 that is not already contained in Figures 1 and 2.

The graphs seem to show that there is a 2-states power-law (breakpoint at 10 un the x-axis) which seems to be consistent with state-of-the-art results. However, to say ””evidenced by the straight line in the log-log plot” is an error. Lines show, they do not evidence, prove nor confirm. The numerical values and statistical tests *do* show and prove. Unfortunately, they are missing in this paper. Unless shown, these results are meaningless.

Related with this discussion, I do not see why the authors did not use this same technique in order to verify that the Heaps’ law is followed as well. That is: why don’t you compare your data with di↵erent distribution as you did to see if Zipf’s law was consistent. This is not what you do for Heaps’ law (as stated in lines 219-229). Why this difference of methodology?

(As an aside comment, lines 219-229 are extreamly confusing and diffcult to follow).

Response: Thank you for your comments. We used the fitting procedure (likelihood, parameter estimation, and model selection) for rank-frequency distributions (i.e., Zipf’s law) because they are probability distributions. In contrast, Heaps’ law is a relationship between the number of words and unique words, which is not a probabilistic distribution. Therefore, we simply used the least squares method for fitting a scaling law with a break point m and calculated R-squared values to confirm the goodness of fit. Following your comments, we have improved the sentence (line 193) and added the distribution of R-squared values as Fig S3.

---- Comment 5 -----

In Section ”Relationship between word production patterns and cognitive scores” authors state that ”we found no significant relationship between the cognitive score and the total number of words spoken (Spearman’s 237 correlation coeffcient p = -0.11, p = 0.38)”. I think that using a correlation test between a discrete variable (cognitive score) and a ”continuous” variable (number of words) is an error. If it is not, please reference a paper where this technique has been used in a similar scenario (discrete vs continuous variable, where the discrete variable has only 5 possible values) Instead of that, there are some other techniques that could be used, as for instance, an Anova test between the alpha average and sd for score -2 and 2 (for instance) and then for scores -1 and 1, -2 and 0 etc... This test would say if the average value for the alpha (beta) values for each score are significantly di↵erent. A correlation does not seem to be convincing at all. It would help also to provide the average + sd for each cognitive score value in a graph. This comment should be extended to the rest of correlations that have been performed with the cognitive score variable and I give them low or little validity.

Response: The cognitive scores we used here were not discrete (-2,-1,0,1,2) but continuous variables (normalized with mean = 0 and S.D. = 1). They were obtained from the first principal component of the four cognitive tests. The values are reflected in continuous color in Figure 1.

---- Comment 6 -----

There is another issue with the relationship between the length of the conversa-

tions/productions and the rest of the analyses. First of all, lines 192-194 show that there

is a large variability in both length and vocabulary. This means that there are speakers that

speak more than others significantly. This induces a bias to the analysis. The authors claim

that there is ”no significant relationship between the cognitive score and the total number

of words spoken” (the correlation is not significant). This seems to indicate that the results

that are later obtanied relating the cognitive scores and the alpha/beta parameters are free of

interference from the length. But this is not necessary true (taking aside the methodological

problem that I just commented in the preceeding paragraph). In order to prove that the length

(word production) is taken out of the equation (that is: length does not bias the analyses) one

should make the analysis taking fixed-length samples and comparing the results obtanied in

the same analyses with thoses presented in the paper. If the results were similar, then, it could

be concluded that text length does not bias the results. Assuming that this e↵ect may be

”cancelled” by the fact that there is no relation between two of the variables is not necessarily

Response: Thank you for your comments. We have already performed the analysis with a fixed word length. Fig 4 shows that when the number of words is roughly larger than 10,000, a significant relationship between them can emerge. We have improved the explanation of the analysis (lines 235-250).

---- Comment 7 -----

 The authors also state that ”the longer the data length (e.g., > 10,000), the higher the corre- lation coeffcient (e.g., r close to 0.5) between the Heaps’ exponent and the cognitive score”. That is: this implies that the longer the conversation, the more correlated the heaps exponent to the cognitive score, then, it implies that the length is relevant to this analysis, that is, the length *biases* the analysis. Said otherwise: we could have that it is the length of the text that provides a higher value of the parameter. This seems inconsistent with the above paragraph.

Response: As mentioned above, figure 3 shows the result of the analysis with a fixed word length. We have improved the explanation of the analysis (lines 235-250).

---- Comment 8 -----

But moreover, this would not mean that ”it is not necessary to analyze datasets with tens of

thousands of words for each participant”, as the authors point out. Maybe, a longer length

would exhibit different results. It just means that for the lengths that are in the dataset, it is

not relevant.

Response: Thank you for your comment. We have removed the sentence.

----- Comment 9 -----

As for the section ”Computational models bridging scaling laws and cognitive functions”, I have not been able to understand the meaning of this section in this paper. The material contained here seems completly irrelevant to me, but I may be wrong. I would appreciate a reason why this section o↵ers a new insight or adds new ideas. It seems that the authors tested a theoretical models, but I do not see any relationship with the rest of the paper.

Response: Thank you for your comment. We totally agree. Hence, we removed the computational model section.

---- Comment 10 -----

 In Section ”Source of new words”, gives me some deep theoretical doubts. The authors seem to suggest that there is a way to find out what words uttered by the speakers have been acquired during the conversations and which belonged already to the speaker’s repertoire. However, it seems to me very unlikely to assume that old people acquire new words at all. Therefore, the assumption in this section seems seriously flawed to me: what is the evidence that this feature (word acquisition) is a productive feature in aged people? It would make perfectly sense in the case of a longitudinal study of children that are acquiring words, but in this case, it seems tremendously unclear and it has no theoretical support (at least, I see no reference in the text). If a subject utters a word in the conversation, it may be that this is because he heard it from a conversation partner. But from here, one can’t imply that the subject *acquired* that word taking into account the age of the individuals analyzed (here I mean *acquire* in the linguistic sense of adding it to his own lexicon).

Response: Thank you for your comments. We apologize for the confusion. ”New words” in our manuscript do not mean that elderly people acquired new words into their lexicon. Instead, we mean ”newly appeared words in each person’s conversational data”. For example, at the beginning, person A said ”I like apples”, and then person B replied ”I like apples too!” In this case, person A produced a new word ”apples” followed by person B. The appearance of new words suggests that the words transfer from one person to another or a person spontaneously speaks the word. To avoid confusion, we improved the explanation (lines 262-264).

---- Comment 11 -----

DISCUSSION. In the Discussion section, the authors claim that ”In contrast, MCI or dementia patients might not follow the scaling laws because repeating a certain word due to critical cognitive impairment or memory disorder may result in the collapse of the scaling laws”. This is just a speculation that has no empirical evidence in the paper.

Response: Thank you for your comment. We have removed the sentence.

---- Comment 12 -----

This section (Discussion) is a mix of di↵erent subjects with no meaningful narrative. I strongly suggest to rewrite this section, following a coherent line of discussion, not mixing future work (349-360), speculations with the discussion of the material contained on the paper. I would also avoid sentences like ”we revealed the association between cognitive functions and word production patterns” because as this reviewer states above, this is debatable.

Response: Thank you for your comments. We followed up your advice and improved this section (lines 289-303, 311-320).

---- Comment 13 -----

You say that ”While we did not find a significant relationship between Zipf’s exponent and the cognitive score (Fig 2A), the exponents of Heaps’ law, that is, the slope of the relationship between the number of words and di↵erent words, were significantly associated with cognitive function scores”. Taking aside that (as I previously commented on) the methodology that you use to assume that there is a relationship between the exponents of the Heaps’ law and the congitive scores is flawed, the fact that this relation exists, but the relation between exponents of the Zipf’s law and the same scores, how does it relate to the fact that both Zipf’s and Heaps’ laws are related? (see bibliographical suggestions at the end of this text). Is it consistent?

Response: Thank you for your comments. There was a correlation between Zipf’s exponent and Heaps’ exponents in our result. We have added the results and discussions (Fig 2, lines 205-211, 289-303).

---- Comment 14 -----

FINAL REMARKS.

The authors should discuss the relationship between both Zipf’s and Heaps’ laws. Some refer- ences that may be of help: https://journals.plos.org/plosone/article?id=10.1371/journal.pone.0014139
https://doi.org/10.1076/jqul.8.3.165.4101
https://journals.aps.org/prl/abstract/10.1103/PhysRevLett.114.238701

In fact, along the paper, they seem to ignore that both laws are related, and that some of the results could be inconsistent with this assumption. That should also be discussed.

There is a lot to improve the paper, starting by the presentation. However, I would like to say that the material in this paper seems to be of interest. The study of Zipf’s and Heaps’ laws in diferent kind of people and languages, is an interesting topic of research.

But there is a lot to be improved: presentation, motivation, relevant methodological problems, a more systematic presentation of the results, stating *clearly* each of the relationships that have been studied, with what methodology, and present the *full* results (in the appendix if they are too extensive) with figures that help to understand them. Therefore, I would encourage the authors to improve the quality of their paper, be more consistent in their analyses.

Response: We appreciate all your comments. We added the results and the discussion about the relationship between the exponents of Zipf’s and Heaps’ law (Fig 2, lines 205-211, 289-303). We have also added the literature you suggested.

Answers to Reviewer 2

In response to the valuable comments of the referee, I have modified my paper appropriately to address his concerns. Below I give a point-by-point reply to each criticism. Please note that the improved parts in the manuscript are indicated in blue.

---- Comment 1 -----

I appreciate that the authors share the cognitive test results and fitted Heaps/Zipf’s exponents. However, for the sake of reproducibility, I would suggest to publish also the processed conversational data. While the fitting method used by the authors seemed robust enough to me, fitting of fat-tailed distributions is a delicate issue and many fitting procedures have been proposed in the literature. Therefore, I would be more comfortable if the reader is given access to the raw tokenized dataset, which would allow him/her to (i) verify the results of the manuscript and (ii) perform additional alternative analysis of the dataset. There are simple solutions to publish large datasets such as Zenodo.

Response: Thank you for your helpful comment. We followed your suggestion and deposited the conversational data on figshare (URL). We have added the description (lines 131-132). Note that because it includes some privacy information, we converted the words into other unique numbers.

---- Comment 2 -----

The R code / necessary scripts to reproduce the results of the paper should also be published (as per PLOS editorial policy).

Response: Thank you for your comments. We followed your advice and attached the R code in the supporting information (lines 132-133).

---- Comment 3 -----

Is there a measurement error associated to the cognitive tests results? For instance, if each subject performed each test more than once, then it would be better to report the variances in addition to the averages. If each test was only taken once then it would be good to know, perhaps from the literature, what are the typical inter-subject variabilities of said tests.

Response: Thank you for your comments. We have accordingly added a description of the test-retest reliability for each cognitive test from the literature (lines 98-102).

---- Comment 4 -----

 I am not convinced the PCA-based cognitive score is a good metric for several reasons. First, it only captures 40% of the variability. My understanding is that the di↵erent cognitive tests measure di↵erent cognitive functions which could but do not necessarily correlate with each other. If that is the case, then taking the PCA of all measures might obscure more interesting results. Second, using the first principal component as a “summary” cognitive score has the undesirable consequence that scores of different subjects are not independent of each other anymore: to compute the score of one subject, we need the test results of all other subjects. I suggest the authors mention, at least, the loads of the first principal component, so that the reader has an idea of what are the weights of each test in the final cognitive score, and perform some additional analysis based on each cognitive test separately.

Response: Thank you for your helpful comments. We analyzed the relationship between scaling laws and each cognitive score (lines 231-232). The results are shown in Table 3. We also added the factor loading of PC1 (Table 1).

---- Comment 5 -----

Zipf’s law and Heaps’ law are statistical laws tightly dependent on each other, see for instance (Lu ¨, Zhang, and Zhou 2010; van Leijenhorst and van der Weide 2005; Font-Clos and Corral 2015, ). In this sense, the fact that only Heaps’ exponent significantly correlates with cognitive scores –but not Zipf’s exponent— might just be a technical artifact. Regardless, the number of tokens could certainly be a confounding variable in this case: First, there has been already ample debate in the community regarding the relation between text length and Zipf’s exponent, see (Corral and Font-Clos 2017; Bernhardsson, da Rocha, and Minnhagen 2009) (in summary, and depending on the fitting methodology etc, one tends to obtain larger exponents for longer texts). Second, having a quick look at the supplementary CSV data, it would seem that text length is a very good predictor of Heaps’ exponent, but not of cognitive score. The authors might want to attempt to more clearly untangle the role of text length in their analysis.

Response: Thank you for your helpful comments. We totally agree with you. We analyzed the relationship between Zipf’s and Heaps’ exponents. A significant negative correlation between α and β was found (Fig 2, lines 205-211), although Zipf’s exponent was not significantly correlated with cognitive scores. We have added the results and the discussion about this (lines 289-303).

---- Comment 6 -----

The reasoning of lines 269-272 is very unclear to me: basically, the Figure shows that in short datasets the association is lost, which would imply the opposite of what is being said? That is, we need to analyze large datasets to make sure we observe a meaningful association.

Response: Thank you for the comment. We improved the sentence accordingly.

---- Comment 7 -----

I do not see a clear connection between the computational model and brain cognitive functionality. Clearly, if Heaps’ exponent correlates with cognitive score, then this will also be captured by the model of Gerlach et al, but that does not bring new information per se. The authors say: “we focused on a parameter related to the decay rate of probability for new word production and interpreted it as a cognitive function.” So, the relation between word produc- tion (Heaps’ law) and cognitive function is an assumption of the authors, not a conclusion obtained from the analysis. This is clearly seen in Figure 4A (which would be obtained with any other dataset).

Response: Thank you for your comments. We totally agree. We decided to remove the model section from the manuscript.

---- Comment 8 -----

The authors claim “the scaling laws are useful to detect the tendency of cognitive decline, even in healthy people.”. I do not think this conclusion is well supported by the analysis presented in this manuscript.

Response:Thank you for your comments. We have removed the sentence. 

---- Comment 9 -----

The authors claim that “out approach requires only limited data from which to detect the relationship between cognitive functions and word patterns, even for healthy participants.”, at odds with the results presented in Figure 3.

Response: Thank you. Following your comments, we removed the sentence.

---

## [Decision Letter · Decision Letter 1]

8 Dec 2020

PONE-D-20-21971R1

Scaling laws in natural conversations among elderly people

PLOS ONE

Dear Dr. Abe,

Thank you for submitting your manuscript to PLOS ONE. After careful consideration, we feel that it has merit but does not fully meet PLOS ONE’s publication criteria as it currently stands. Therefore, we invite you to submit a revised version of the manuscript that addresses the points raised by reviewer 1. In particular, the criticism of the statistical analysis indicates that some of the conclusions would need to be re-considered, please address all the points and revise your conclusions accordingly as this is expected to be the last round of review.

The reviewer's criticism on PLOS ONE's manuscript format will not be taken into account for the decision to accept the manuscript. Nevertheless, you may want to include a pdf version of your manuscript with figures in place.

We look forward to receiving your revised manuscript.

Kind regards,

Eduardo G. Altmann

Academic Editor

PLOS ONE

Reviewers' comments:

Reviewer's Responses to Questions

**Comments to the Author**

1. If the authors have adequately addressed your comments raised in a previous round of review and you feel that this manuscript is now acceptable for publication, you may indicate that here to bypass the “Comments to the Author” section, enter your conflict of interest statement in the “Confidential to Editor” section, and submit your "Accept" recommendation.

Reviewer #1: All comments have been addressed

Reviewer #2: All comments have been addressed

2. Is the manuscript technically sound, and do the data support the conclusions?

Reviewer #1: Partly

Reviewer #2: (No Response)

3. Has the statistical analysis been performed appropriately and rigorously? 

Reviewer #1: No

Reviewer #2: (No Response)

4. Have the authors made all data underlying the findings in their manuscript fully available?

Reviewer #1: Yes

Reviewer #2: (No Response)

5. Is the manuscript presented in an intelligible fashion and written in standard English?

Reviewer #1: Yes

Reviewer #2: (No Response)

6. Review Comments to the Author

Reviewer #1: The paper is now easier to read and somehow more consistent. However, some serious flaws still persist. I will try and explain it in the following.

Before that, I would like to discuss again the format of the paper. In this version that you have submitted, when a link to a graph is clicked, the focus moves to the caption of the graph, but the graph is somewhere else. It is extremly upsetting to read a paper like this. If in the previous version the problem was that you were using Word instead of latex, I do not see why this still happens when you use latex. As you may know, latex has a way to place the graphs in the pace where they are defined in the .tex file ([h!]). Still worse, the graphs are vertical instead of horizontal. That means that when I have a reference to your graph, I click it, and then, I just see a caption, and then, I need to find (and figure out) where the graph is and which one it is, and after all this, then, I need to rotate the page 90 degrees in order to be able to read it.

We review papers just as a service to the community, and not only for free, but also at the expenses of our efforts and time. This is acceptable, and I do not complain for it, I am happy to help, but I do not accept that I need to read the paper in such an uncomfortable and difficult way, specially when the tools to prevent this mess are easily available. This comment is not only for the authors, but for the editors as well.

Comments:

First of all, at the end of the abstract the authors state that:

"We found that word patterns followed these scaling laws irrespective of cognitive function, and that the variations in Heaps’ law were associated with cognitive function. Moreover, variations in Heaps’ law were associated with the ratio of new words taken from the other participants’ speech. These results indicate that scaling laws in language are related to cognitive processes.".

This whole paragraph is contradictory. They say that "these scaling laws [are] irrespective of cognitive function", this is "these laws" are both Zipf's and Heaps', and then, they say: "variations in Heaps’ law were associated with cognitive function". The following sentence still adds more contradiction to the sentence. In any case, I guess that this is just a writing error.

In line 53, the authors say that "Heaps’ law describes how new words are produced along with sentences or during conversations". This is not the meaning of this law, and I think that the mistake on the interpretation of this law implies more serious problems along the paper. Heaps' law state that the number of different words in a text is a funtion proportional to the length of the text (modulo exponent). This has nothing to do with the idea of "new" words, in the sense of words that did or did not belong to the speaker's lexicon. This law describes how the variety of different words varies when we write or speak. It may seem that the only problem of the interpretation of the meaning of this law is just that the authors use the word "new" where I use the word "different", but I will discuss it later, to show that, from this reviewer's point of view, the authors have mistaken or misused the meaning of this law.

My main concern with the results shown in the paper are related to the Heaps' law results.

In line 227 the authors state that:

"the relationship between the exponent \\beta of Heaps’ law and cognitive scores and found a significant relationship (p = 0.002)".

But there is a value that goes along this p-value, which is 0.003. What is the meaning of this value? Or, said otherwise, the fact of being statistically significant is important, but then, we need to see the slope (in case of a correlation analysis) or an extra metric that describes the nature of this significant p-value. To make it clearer, when you have a significant correlation, then you have a look at the slope, since it is not the same to have a significant correlation of a slope -> 0 than a significant correlation with a slope -> 1/-1. In this case, apart from the significance, what else can be said about the nature of both relations? This is important to clarify because of Zipf's and Heaps' laws are connected, then, the authors need to be very precise when they state that the expected behavior (assuming transitivity between the relations: cognitive score - Zipf's law, cognitive score-Heaps law, Heaps' law - Zipf's law) does not hold according to they results.

In lines 231-234 the authors state that:

"We confirmed a robust relationship between the exponent \\beta and all original cognitive scores, except for the digit span (Table 3)."

(apart from the very liberal use of the word "robust" in this particular case) and then they say:

"Thus, these results indicate that the variation in Heaps’ law could be associated with the difference in cognitive functions."

Yet, you also state that:

1. there is no relationship between cognitive score and number of uttered words (line 217).

2. in figure 4 you show a relationship between cognitive score and exponent \\beta and length of text.

This seems inconsistent, taking into account that transitivity should apply in these cases. I did not find any discussion about this fact in the paper.

Finally, I would like to comment section "Source of new words". After the response to one of my questions, I firmly think that the definition of "new words" that they apply in this paper has nothing to do with the meaning of Heaps' law. What they do is to analyze the relation of different words only if they have been uttered before by another speaker. This is not what the law states, since the law makes no difference about the "origin" of the words or if they were already in the speaker's lexicon or they just learned it. This law measures the proportion of different words w.r.t. number of uttered words. Therefore, selecting only those words that have been uttered by someone else, the authors are biasing the analysis. I do not see any meaning on analyzing only this subset of words.

Moreover, in order to compute the parameter of Heaps' law, it seems clear that the longer the text is, the more accurate the computation of this parameter will be, since this function will have a larger size span to be fitted. And *precisely* because of that, this parameter *needs* to be computed with fixed length text, if the purpose of the analysis is to see relevant differences between speakers' performance. It comes to no surprise to me that the longer the text, the higher the value of this parameter is. In fact, I would say that the longer the text is, the more *accurate* the value of this parameter is (but this is just a guess).

If I had to see if this parameter had an impact or a relationship with the cognitive score, I would take some individuals with a significant low score and some with a significant high score (w.r.t. average, for instance), obtain their uttered words, set a prefix length fair for all of them (the minimum is usually taken), compute the Heaps' parameter for all of them, and then, apply a method to see if the difference (in average, for instance) is significant. Or you could group them as well (low, medium, high cognitive score), and see the statistical differences between all of them. Using you methodology, you make a mistake (from my point of view):

1. using the words that you define as "new".

2. not setting a prefix length.

3. not using more precise and clearer (and yet simple) statistical tools to find out the relation between individuals.

In is not enough to see if there is a relation between the cognitive score and the length, because not taking a prefix to compute Heaps' parameter is biased by you decision of taking a fixed prefix length.

In fact, the last graph may mean nothing, since you are not using all the available utterances for the analysis, and the graph B in the previous page may just mean that the more words you take, the more precise is your computation of the Heaps' parameter.

That means that your statement in lines 298-299 is dubious.

Reviewer #2: All comments have been addressed.

7. PLOS authors have the option to publish the peer review history of their article (what does this mean?). If published, this will include your full peer review and any attached files.

Reviewer #1: No

Reviewer #2: **Yes: **Francesc Font-Clos

---

## [Author Response · Author response to Decision Letter 1]

15 Jan 2021

Comment1:

Before that, I would like to discuss again the format of the paper. In this version that you have submitted, when a link to a graph is clicked, the focus moves to the caption of the graph, but the graph is somewhere else. It is extremly upsetting to read a paper like this. If in the previous version the problem was that you were using Word instead of latex, I do not see why this still happens when you use latex. As you may know, latex has a way to place the graphs in the pace where they are defined in the .tex file ([h!]). Still worse, the graphs are vertical instead of horizontal. That means that when I have a reference to your graph, I click it, and then, I just see a caption, and then, I need to find (and figure out) where the graph is and which one it is, and after all this, then, I need to rotate the page 90 degrees in order to be able to read it.

We review papers just as a service to the community, and not only for free, but also at the expenses of our efforts and time. This is acceptable, and I do not complain for it, I am happy to help, but I do not accept that I need to read the paper in such an uncomfortable and difficult way, specially when the tools to prevent this mess are easily available. This comment is not only for the authors, but for the editors as well.

Response:

We are sorry for making you uncomfortable again. We understand what you mean. However, this is a PLOS ONE's problem. The position of the graphs must be at the end of the manuscript even for Latex format, according to the submission guidelines of PLOS ONE.

The guideline also says that clicking the figure references in the text should take you to not the figures, but the caption. The Latex template of PLOS ONE is designed to do so. Thus, we followed these guidelines.

Comment2:

First of all, at the end of the abstract the authors state that:

"We found that word patterns followed these scaling laws irrespective of cognitive function, and that the variations in Heaps’ law were associated with cognitive function. Moreover, variations in Heaps’ law were associated with the ratio of new words taken from the other participants’ speech. These results indicate that scaling laws in language are related to cognitive processes.".

This whole paragraph is contradictory. They say that "these scaling laws [are] irrespective of cognitive function", this is "these laws" are both Zipf's and Heaps', and then, they say: "variations in Heaps’ law were associated with cognitive function". The following sentence still adds more contradiction to the sentence. In any case, I guess that this is just a writing error.

Response:

Thank you for your comments. We considered two points about scaling laws: 1) whether the participants exhibit scaling laws and 2) if they exhibit scaling laws, is there a variation in exponents? And are these exponents related to cognitive functions? These raised problems are written in Introduction (lines 62-69).

For the first question, we analyzed the rank-frequency relationship of words and the number of words and different words relationship and evaluated the goodness-of-fit. We confirmed that all they commonly exhibited two scaling laws. Therefore, this point is not irrespective of cognitive functions.

For the second question, we found the variations in scaling exponents and the significant relationship between the exponents $\\beta$ and cognitive functions. Therefore, although whether words patterns follow scaling laws or not is irrespective of cognitive scores, the variations in the exponent $\\beta$ is associated with cognitive scores.

To avoid confusion, we improved the abstract.

Comment3:

In line 53, the authors say that "Heaps’ law describes how new words are produced along with sentences or during conversations". This is not the meaning of this law, and I think that the mistake on the interpretation of this law implies more serious problems along the paper. Heaps' law state that the number of different words in a text is a funtion proportional to the length of the text (modulo exponent). This has nothing to do with the idea of "new" words, in the sense of words that did or did not belong to the speaker's lexicon. This law describes how the variety of different words varies when we write or speak. It may seem that the only problem of the interpretation of the meaning of this law is just that the authors use the word "new" where I use the word "different", but I will discuss it later, to show that, from this reviewer's point of view, the authors have mistaken or misused the meaning of this law.

Response:

Thank you for your comments. In our paper, we use "new words" as newly-used words (or newly-appeared words) in each participant's data set. We do not mean that new words are added to his/her vocabulary or lexicon. Such a way to use "new words" appeared in previous studies (e.g., Gerlach and Altmann 2013). Anyway, to avoid confusion, we removed the sentence in line 53 of the previous manuscript and changed "new words" to "newly-used words" or "different words" in a whole manuscript.

Comment4:

My main concern with the results shown in the paper are related to the Heaps' law results.

In line 227 the authors state that:

"the relationship between the exponent $\\beta$ of Heaps’ law and cognitive scores and found a significant relationship (p = 0.002)".

But there is a value that goes along this p-value, which is 0.003. What is the meaning of this value? Or, said otherwise, the fact of being statistically significant is important, but then, we need to see the slope (in case of a correlation analysis) or an extra metric that describes the nature of this significant p-value. To make it clearer, when you have a significant correlation, then you have a look at the slope, since it is not the same to have a significant correlation of a slope -> 0 than a significant correlation with a slope -> 1/-1. In this case, apart from the significance, what else can be said about the nature of both relations? This is important to clarify because of Zipf's and Heaps' laws are connected, then, the authors need to be very precise when they state that the expected behavior (assuming transitivity between the relations: cognitive score - Zipf's law, cognitive score-Heaps law, Heaps' law - Zipf's law) does not hold according to they results.

Response:

Table 2 shows the statistical result of the linear mixed model, including the slopes.

"Explanatory variable: estimates (SE, p-value)" in the upper of Table 2 indicates that values in each cell represent the estimated regression coefficient, the standard error of the coefficient, and the p-value. For example, the regression coefficient, the standard error, and the p-value in cognitive score vs $\\beta$ are 0.009, 0.003, and 0.002, respectively. The result indicates that as the cognitive scores increases by 1, beta increases by 0.009.

Also, there are inevitable random factors in the relationship between variables.

Hence, transitivity does not always hold. We discussed the relationship among $\\alpha$, $\\beta$, and cognitive functions (lines 297-306).

Comment5:

In lines 231-234 the authors state that:

"We confirmed a robust relationship between the exponent $\\beta$ and all original cognitive scores, except for the digit span (Table 3)."

(apart from the very liberal use of the word "robust" in this particular case) and then they say:

"Thus, these results indicate that the variation in Heaps’ law could be associated with the difference in cognitive functions."

Yet, you also state that:

1. there is no relationship between cognitive score and number of uttered words (line 217).

2. in figure 4 you show a relationship between cognitive score and exponent $\\beta$ and length of text.

This seems inconsistent, taking into account that transitivity should apply in these cases. I did not find any discussion about this fact in the paper.

Response:

Thank you for your comments.

We meant no significant correlation (calculated by Spearman's correlation coefficient) between the number of words and the cognitive function scores. To avoid confusion, we improved the sentence (line 217).

In Figure 4, the x-axis is the fixed word length, and the y-axis is the correlation coefficient between cognitive scores and $\\beta$.

It suggests that many words make the relationship between cognitive scores and $\\beta$ clear. Therefore, it does not indicate that participants with a large number of words have high cognitive scores. Thus, it is not related to the transitive relationship.

Comment6:

Finally, I would like to comment section "Source of new words". After the response to one of my questions, I firmly think that the definition of "new words" that they apply in this paper has nothing to do with the meaning of Heaps' law. What they do is to analyze the relation of different words only if they have been uttered before by another speaker. This is not what the law states, since the law makes no difference about the "origin" of the words or if they were already in the speaker's lexicon or they just learned it. This law measures the proportion of different words w.r.t. number of uttered words. Therefore, selecting only those words that have been uttered by someone else, the authors are biasing the analysis. I do not see any meaning on analyzing only this subset of words.

Moreover, in order to compute the parameter of Heaps' law, it seems clear that the longer the text is, the more accurate the computation of this parameter will be, since this function will have a larger size span to be fitted. And *precisely* because of that, this parameter *needs* to be computed with fixed length text, if the purpose of the analysis is to see relevant differences between speakers' performance. It comes to no surprise to me that the longer the text, the higher the value of this parameter is. In fact, I would say that the longer the text is, the more *accurate* the value of this parameter is (but this is just a guess).

If I had to see if this parameter had an impact or a relationship with the cognitive score, I would take some individuals with a significant low score and some with a significant high score (w.r.t. average, for instance), obtain their uttered words, set a prefix length fair for all of them (the minimum is usually taken), compute the Heaps' parameter for all of them, and then, apply a method to see if the difference (in average, for instance) is significant. Or you could group them as well (low, medium, high cognitive score), and see the statistical differences between all of them. Using you methodology, you make a mistake (from my point of view):

1. using the words that you define as "new".

2. not setting a prefix length.

3. not using more precise and clearer (and yet simple) statistical tools to find out the relation between individuals.

In is not enough to see if there is a relation between the cognitive score and the length, because not taking a prefix to compute Heaps' parameter is biased by you decision of taking a fixed prefix length.

In fact, the last graph may mean nothing, since you are not using all the available utterances for the analysis, and the graph B in the previous page may just mean that the more words you take, the more precise is your computation of the Heaps' parameter.

That means that your statement in lines 298-299 is dubious.

Response:

Although you may misunderstand our analysis, we analyzed all available words in the conversation for scaling laws, not limited to words uttered before by another speaker. In the section of Source of new words, we analyzed the origin of newly-used words in each participant. 

It can be independent of scaling laws. 

Also, we already showed the result of the fixed-length data for scaling laws in Figure 4.

As for prefix, we analyzed data without prefix. However, the results almost never changed (lines 234-236). We added them to Supporting Information (Appendix S3, S4-S8 Figs, and S2-S3 Tables). Additionally, we found a mistake in a caption of Fig. 5 and improved it.

---

## [Decision Letter · Decision Letter 2]

28 Jan 2021

Scaling laws in natural conversations among elderly people

PONE-D-20-21971R2

Dear Dr. Abe,

We’re pleased to inform you that your manuscript has been judged scientifically suitable for publication and will be formally accepted for publication once it meets all outstanding technical requirements.

Kind regards,

Eduardo G. Altmann

Academic Editor

PLOS ONE

Additional Editor Comments (optional):

Reviewers' comments:

Reviewer's Responses to Questions

**Comments to the Author**

1. If the authors have adequately addressed your comments raised in a previous round of review and you feel that this manuscript is now acceptable for publication, you may indicate that here to bypass the “Comments to the Author” section, enter your conflict of interest statement in the “Confidential to Editor” section, and submit your "Accept" recommendation.

Reviewer #1: All comments have been addressed

2. Is the manuscript technically sound, and do the data support the conclusions?

Reviewer #1: Yes

3. Has the statistical analysis been performed appropriately and rigorously? 

Reviewer #1: Yes

4. Have the authors made all data underlying the findings in their manuscript fully available?

Reviewer #1: Yes

5. Is the manuscript presented in an intelligible fashion and written in standard English?

Reviewer #1: Yes

6. Review Comments to the Author

Reviewer #1: I apologize for blaming the authors and the editor for the poor presentation, which is solely the responsibility of the journal. I hope that this can be improved in the future, taking into account that, as I previously said, the reviewers are working for free for the journal.

7. PLOS authors have the option to publish the peer review history of their article (what does this mean?). If published, this will include your full peer review and any attached files.

Reviewer #1: No

---

## [Editor Report · Acceptance letter]

1 Feb 2021

PONE-D-20-21971R2 

Scaling laws in natural conversations among elderly people 

Dear Dr. Abe:

I'm pleased to inform you that your manuscript has been deemed suitable for publication in PLOS ONE. Congratulations! Your manuscript is now with our production department. 

Kind regards, 

on behalf of

Dr. Eduardo G. Altmann 

Academic Editor

PLOS ONE